# Nanomedicine for Cancer and Autoimmune Immunotherapy

**DOI:** 10.3390/ijms262411941

**Published:** 2025-12-11

**Authors:** Ashi Ramavat, Priya Antil, Soniya Kaushik, Baby Kataria, Ramendra Pati Pandey

**Affiliations:** 1Department of Biotechnology, SRM University, Delhi-NCR, Sonepat 131029, Haryana, India; ashi8472@yahoo.com (A.R.);; 2Department of Biotechnology, Deenbandhu Chhotu Ram University of Science and Technology (DCRUST), Sonepat 131001, Haryana, India; 3Department of Microbiology, SRM University, Delhi-NCR, Sonepat 131029, Haryana, India

**Keywords:** tumor microenvironment (TME), CAR-T cell enhancement, cytokine regulation, exosome-mimetic nanoplatforms

## Abstract

Nanomedicine has now become a transformative platform that enhances the precision and efficacy of immunotherapy approaches and allows customizations like never before when it comes to cancer, as well as autoimmune conditions. Using platforms based on nanoscale, researchers have been able to manipulate immune responses operating across spatial and temporal scales to address key limitations of conventional immunotherapy associated with working with immune response such as immune evasion, systemic toxicity, and poor pharmacokinetics. Sophisticated nanoparticles (such as stimuli-sensitive ones, exosome-mimetic vesicle nanoparticles, and nanoparticles with CRISPR) allow directed immunomodulators, antigens, and gene-editing systems to reach one or more particular immune compartments. The innovations allow reprogramming of immune cells, immune tolerance rejuvenation, and expansion of antitumor immunity without significant off-target effects. Finding applications in integrating the artificial intelligence as well as multi-omics techniques, the process leads to personalization of the nano-immunotherapies based on patient-specific immuno-signatures. The chapter discusses the mechanistic rationale, therapeutic advancement, and the translational opportunities of nanotechnology-based immunotherapies that define them as part of a foundation of future generations of clinical approaches to precision immune modulation in oncology and autoimmune diseases.

## 1. Introduction

With the introduction of nanotechnology into the medical field, nanomedicine as a combination of these two disciplines has become a ground-breaking approach in the treatment of cancer and immunology with new solutions offered in the field diagnosis, targeted therapy, and prevention of the disease. Nanomedicine can take special advantage of exploiting nanomaterials (scale 1–100 nm), providing strengths such as enhanced penetration to tissues and therapeutic agents, controlled drug delivery, and personalized immune response interactions. Such properties are especially advantageous in the field of immunotherapy, where a fine balance between control over the immune response and favorable versus adverse effects is important to therapeutic efficacy, having the benefits of high specificity [1]. Along with a growing understanding of immune derangement in cancer and autoimmune diseases, the role of nanomedicine in immunotherapy has grown. The two types of diseases, although being different in their clinical presentation, are based on a common mechanism of immune disturbance. One of the recognized ways is that the evasion of immune response in cancer cells enhances the growth and development of cancerous cells through immune evasion via down-regulation of immune checkpoints (e.g., PD-L1), recruitment of immunosuppressive cells (e.g., Tregs, MDSCs), and release of inhibitory cytokines into the microenvironment of the tumor (TME). Autoimmune diseases in contrast are caused by loss of self-tolerance, pathologic activation of autoreactive B and T lymphocytes, unresolving inflammation, and destruction. Such complexities explain the need to develop therapeutics that can be used to fine-tune immune response and thereby reinstate surveillance in oncology or re-induce to tolerance in autoimmunity [2].

The development of nanomedicine in this respect represents a gradual trend towards highly complex, immune-responsive platforms starting with the passive drug vehicles. The first nanomedicines approved clinically, namely liposomal doxorubicin (Doxil^®^), were manufactured in such a way that they would maximize pharmacokinetics and minimize the toxicity of the chemotherapeutic drugs. Nevertheless, the subject area has gone beyond passive delivery means to target immune processes directly [3]. The use of nanoparticles in oncology is emerging in the context of targeted delivery to tumors of immunostimulatory targets (tumor antigens, adjuvants, cytokines, monoclonal antibodies, and nucleic acids) to augment tumor immunogenicity, address immune exhaustion, and augment the effectiveness of therapies including immune checkpoint inhibitors and adoptive T-cell transfer. In the autoimmune diseases, nanomedicine efforts have concentrated on reinstating immune tolerance, specifically directing immune-cell subsets and inflamed tissues in a selective manner. Tolerogenic antigens, immunosuppressive medications, or genetic endowments can be conveyed by nanoparticles and selectively quiet down pernicious immune reactions without disturbing framework immunity. These kinds of accuracy diminish the wide immunosuppression and the riskiness usually observed in the conventional treatment. As an example, CD4^+^ T-cells responder tolerance to autoantigens delivered to them by polymeric nanoparticles in a controlled-release form has been demonstrated in preclinical models of multiple sclerosis and type 1 diabetes [4].

Nanomedicine can serve as immunomodulatory since it has the ability to provide specific, regulated, and multi-functional delivery alternatives. In nanoparticle design, surface ligands enable cell targeted delivery (e.g., dendritic cells, macrophages, T cells), stimuli-response materials to permit on-demand release (pH sensitive, enzyme sensitive), and co-delivery of multiple agents as part of a single constrict (e.g., antigens + adjuvants, siRNAs + checkpoint inhibitors). This flexibility allows this kind of modulation of the immune pathways with more precision as compared to standard methods [5]. In addition, the development of biomimetic nanotechnology, e.g., nanoparticles covered with membranes of immune cells, not only increases biological compatibility, target specificity, and efficacy of therapy but also makes use of natural immunity.

Recent studies indicate the possibility of nanoparticles to transform the immune environment. In the oncology field, nanoparticles have been used to counter immune suppression that occurs in the TME, improve antigen presentation, and boost the impacts of the checkpoint blockade treatment method. In the same way, nanoparticles were found to successfully induce regulatory T cell responses and the reestablishment of immune tolerance in autoimmune models as compared to low system toxicity. The success of lipid-based nanoparticles to deliver mRNA shown by COVID-19 vaccines has also confirmed the possible translational potential of these technologies and generated wider interest in using lipid-based nanoparticles to modulate the immune system (Figure 1) [6].

Thus, nanomedicine is a potential area in immunotherapy providing new approaches to solving the complexity of immune imbalances in carcinomas and autoimmune diseases. Nanotechnology offers a powerful tool in the development of effective, safe, and patient-tailored therapies of the future by combining attributes to target delivery. With controlled research, nanomedicine will assume a central role in influencing future precision immunotherapy.

## 2. Immune Dysregulation in Diseases Pathogenesis

The immune system is essentially put together to defend the host by means of identifying and destroying ailment-causing pathogens and employing abnormal cells such as changed cancerous cells while maintaining tolerance to self-antigens. Immune dysregulation can arise when these tight balancing systems go awry, either delivering immune subversion and the generation of cancer or the loss of self-tolerance that generates autoimmune illness [7]. They tend to be entangled with chronic inflammation, aberrant cytokine signaling, and disruption of communication between the innate and the adaptive arm of immunity. Knowledge about these mechanisms is vital in further development of nanomedicine-based interventions to accomplish the correction of immune dysfunction in cancer as well as in autoimmune disorders.

### 2.1. Immune Surveillance in Cancer: Evasion Mechanisms

Immunologic surveillance implies that the immune system carries out unremitting checks to recognize and destroy early cancerous cells, prior to them developing and forming tumors. Nonetheless, the cancer cells advance intricate schemes to avoid identification and elimination through the immune system causing them to stay, propagate, and spread. The changes in the antigen presentation pathways are one of the main processes. In tumor cells, down-regulation of major histocompatibility complex (MHC) class 1 molecules is common, and they become incompetent to be recognized by claim T lymphocytes (CTLs). Recent studies have demonstrated that manipulation of proteins like 2-microglobin results in improper expression of the MHC-1 and has direct correlation with immune checkpoint blockade resistance, thus representing the importance of antigen presentation in immune evasion [8].

Moreover, tumor microenvironment (TME) is a very critical component in immune responses. Of note, tumor capacity to shape TME by producing a variety of cytokines, including TGF-8, IL-10, and VEGF, and attracting immunoregulatory cells, including regulatory T cells (Tregs), myeloid-derived suppressor cells (MDSCs), and tumor-associated macrophages (TAMs) is central to the regulation of TME [2]. This immunosuppressive microenvironment does not allow activation and the effector immune cells, particularly the CTLs and the natural killer (NK) cells. Moreover, an upregulation of immune checkpoint receptors by tumors cells, e.g., PD-L1 and by T cells; PD-1 also creates a state of T cell exhaustion in which already established regions of inhibitory receptors are the results of T cell exhaustion. The recent discovery of immune checkpoints such as LAG-3 or TIM-3 has also found these checkpoints to play a central role in immune escape and is under intense clinical trials. There are opportunities for enabling the countering of these evasion mechanisms through nanomedicine. Nanocarriers can be designed in such a way that they present immune-modulating compounds within the TME that increase antigen presentation and that overturn immunosuppression. The recent development is lipid-based nanoparticles that co-deliver siRNA against PD-L1 and chemotherapeutics, trigger effective CTLs reactivation, and strengthen anticancer immunity [9].

### 2.2. Autoimmune Diseases: Breakdown of Self-Tolerance Mechanism

The causes or rather mechanisms that cause autoimmune illness are failures of the various mechanisms of the body that turn to defend against the self, allowing the immune system to fight itself. These are systems that work on central as well as peripheral levels. During the development of the lymphocytes in the thymus and bone marrow, central tolerance is created and autoreactive T and B cells are destroyed. The peripheral tolerance mechanisms also inhibit immune response of the mature lymphocyte through the mechanisms of anergy, deletion, and suppression by the regulatory T cells. A breakdown in these checkpoints results in survival and activation of autoreactive lymphocytes, resulting in autoimmune pathology. Autoimmunity relates to the genetic predisposition to a large extent. Diseases predisposition is conferred by polymorphism in genes coding MHC molecules, e.g., HLA-DRB1 in rheumatoid arthritis or immune regulatory proteins such as CTLA-4 and PTPN22 [10]. Nonetheless, environmental factors’ role is also present through molecular similarities (mimicry) and by-stander activation. An example is infections that may cause autoimmune responses when microbial antigens resemble the structure of self-antigens and thus cross-react. There is ample evidence associated with regard to the role of Epstein–Barr Virus (EBV) and multiple sclerosis through the means of molecular mimicry where some studies have noted this relationship in Science. Such effects are further worsened by epigenetic changes such as aberrant DNA methylation and histone acetylation that change gene expression profiles of immune cells to reinforce autoreactivity [11].

Nanomedicine approaches autoimmune disorders to aim at recreating immune tolerance. Tolerogenic nanoparticles to present self-antigens have had potential in preclinical models. These nanoparticles confer a novel specific induction of antigen-specific regulatory T cells and hence dampen autoimmune inflammation without immunosuppressing more globally. To illustrate, nanoparticles of poly(lactic-co-glycolic acid) (PLGA) composed of disease-relevant peptides were shown to be capable of preventing the onset and progression of disease in murine models of type 1 diabetes and multiple sclerosis [12].

### 2.3. Role of Chronic Inflammation and Cytokine Networks

Chronic inflammation is a signature of cancer as well as autoimmune disorders, with opposing results, however. Chronic inflammation in the disease environment of cancer creates a favorable environment of tumor initiation, development, and metastases. Angiogenesis is enhanced whereby pro-inflammatory mediators activate angiogenesis, inhibit anti-tumor immune responses, and enable genetic mutations by attenuating the synthesis of reactive oxygen and nitrogen species. In autoimmune diseases, on the contrary, chronic inflammation continues tissues destruction and enhances the autoreactive immune responses.

At the center of those mechanisms stand cytokine networks that are dysregulated. There is a dual role of pro-inflammatory cytokines like IL-6, TNF-α, and IFN-γ in immunity and pathology. Another example is IL-6 that advances cancer cell survival, production, and immune resistance through the activation of STAT3 signaling routes and elicits persistent inflammation and autoimmunity by overproducing IL-6 in autoimmune disorders. The TNF-α is an important inflammatory factor used in the progression of cancer and is involved in tissue destruction of the autoimmune system. Although necessary to mediate anti-tumor immunity, IFN-γ enhances autoimmune disorders by stimulating MHC molecules and autoreactive lymphocytes activation [13].

Redundancy and pleiotriodity of cytokine networks add to the complexity of cytokine networks, causing the existence of complex feedback loops which maintain pathological conditions. Single-cell transcriptomic studies have given us clues of the heterogenicity of the cellular origins of these cytokines in diseased tissues, which also have context-specific effects in the critical therapeutic targeting. As an example, some researchers used single-cell RNA sequencing to introduce cell subsets which produce cytokines in inflamed tissues, which is promising as the focus of precision therapy. A new tantamount to regulating these networks is proposed with nanomedicine. Systemic side effects could be reduced with nanocarriers that deliver cytokine-neutralizing agents or anti-inflammatory drugs to the affected tissues, besides increasing therapeutic targeting. Dexamethasone or other immunomodulatory agents in liposomal formulations are being tested as able to reprogram inflammatory microenvironments and bring homeostasis to the immune system [14].

### 2.4. Crosstalk Between Innate and Adaptive Immunity in Pathogenesis

Adaptive and innate immunity cross-communication play a crucial role with regard to maintenance of immune homeostasis as well as dynamics of the immune-mediated disease occurrence. There are pattern recognition receptors (PRRRs) (also called Toll-like receptors (TLRs)) of the innate immune cells that realize the pathogen-associated molecular patterns and elicit the inflammatory responses that can affect the adaptive immunity. Nevertheless, the impairment of said interaction may result in persistent inflammation, autoimmune disorders, or tumor immune escape. The cellular principle of this crosstalk revolves around innate immune cells, most importantly dendritic cells and macrophages [15]. A major example is dendritic cell which has some ability to bridge both innate and adaptive immunity. The maturation or the antigen presentation of dendritic cells may be dysfunctional across the tumor specifically to become an anti-generator of autoreactive T cells. Polarization of macrophages is another example of this complexity. M1 macrophages that can activate and stimulate inflammation and anti-tumor responses are classically activated, unlike M2 that inhibit the same inflammation and anti-tumor responses, encouraging tumor growth and being pro-inflammatory, thus inhibiting immunity. Such imbalance in macrophage subsets contributes towards the occurrence of chronic inflammation and tissue destruction in cases of the autoimmune diseases [16].

These innate signals play an important role in the adaptive immune response. Autoreactive T and B cells were maintained by long-term innate immune activation, including antigen presentation and cytokine production. In cancer, the interactions are used to establish an immune-suppressive microenvironment, which favors tumor survival and growth. In the case of autoimmune diseases, the reverse in these pathways will cause maintenance of pathogenic lymphocytes.

Nanomedicines capable of targeting those pathways are actively developed. Nanocarriers may be designed to either adjust TLR signaling, re-program macrophage polarization, or target specific populations of immune cells with immune-regulatory molecules. The example provided of the biomimetic nanoparticles covered in the leukocyte membranes is an illustration of a novel approach to delivering the therapeutic agent to the loci of immune dysregulation and avoiding accidents with its clearance by the mononuclear phagocytic system [17].

Figure 2 demonstrates that nanomedicine regulates the bidirectional crosstalk between innate and adaptive immunity in the pathogenesis of disease. Illustrating the interactions among the macrophages, dendritic cells, T cells, and antibody-producing cells visually, the figure highlights how nanoparticle based interventions can modulate the antigen presentation, activation of immune responses, and the subsequent pathological events.

## 3. Limitations of Conventional Immunotherapies

Immunotherapy is one of the breakthroughs in curing cancer and autoimmune diseases because of the use of the immune response of the body as a clinical tool. Nevertheless, traditional immunotherapeutic approaches suffer immense setbacks notwithstanding their effectiveness in a variety of malignancies and autoimmune diseases. Such constraints interfere with the therapeutic efficacy, reduce the number of patients who can qualify to receive the therapy, and in most instances impose treatment-related adverse effects. The understanding of such limitations is needed in great detail in order to justify the incorporation of nanomedicine in immunotherapy with a greater amount of precision, efficacy, and safety.

### 3.1. Non-Specific Immune Activation and Systemic Toxicity

The greatest shortcoming of most available immunotherapeutic options, especially those focusing on immune checkpoint (ICIs), as well as cytokine-based treatments, is that these modalities are unable to launch an immune response with specificity. The agents also tend to trigger wide-ranging immune reactions that fail to target specific pathological processes, triggering systemic immune-related adverse events (irAEs). As an example, ICIs against CTLA-4 or PD-1/PD-L1 axis may cause autoimmune-like toxicity of the skin, gastrointestinal tract, endocrine glands, and lungs [18]. These kinds of side effects do not only reduce the quality of life of a patient but may also lead to the termination of treatment or to immunosuppressive procedures that dull the effects of the treatment.

The potential danger of systemic toxicity is seen in such cytokine-based therapy as high dose interleukin-2 (IL-2) given to metastatic melanoma or renal cell carcinoma. Although IL-2 is effective in activating potent antitumor responses of T cells, cytokine causes vascular leak syndrome, severe hypotension, and multi-organ dysfunction through the non-specific activation of immune systems. Likewise, in autoimmune diseases, general immunosuppressants lack specificity, thus leading to overall immune suppression leading to making one prone to infections and cancers [19]. The lack of traditional immunotherapies leads to the necessity to develop targeted transport agents, like nanocarriers, which can localize the immune modulators to a particular tissue, a cell, or even microenvironment, ultimately leading to enhanced specificity of a therapeutic approach whilst reducing systemic exposure.

### 3.2. Limited Tumor Penetration and Off-Target Effects, and Challenges in Therapeutic Durability

A major drawback with regard to traditional immunotherapies is their slight tumor penetrability, bad tissue specificity, and failure to be therapeutically durable. Physical and biological barriers caused by the dense extracellular matrix, aberrant vasculature, and hypoxic conditions present in solid tumor thus make it difficult to infiltrate immune cells and therapeutic agents effectively. This results in unequal distribution in the tumor and therefore low eradication and remission of the tumor. Even more, resistant to the current immune checkpoint blockade treatment are tumors that have poor immune infiltration, also known as cold tumors.

Another way by which an increase in off-target effects is made through the use of immunotherapeutic agents by systemic administration. Cytokine and monoclonal antibodies, even when they are targeted at a particular molecule, could bind with health tissues that also produce low amounts of the intended molecules. This may cause unexpected damage to any tissues together with the numerous side effects, including severe ones like cardiotoxicity or immune related events, which makes the treatment even more complicated. Moreover, cancer and autoimmune diseases are both characterized by resistance-developing mechanisms that compromise long-term effectiveness [20]. Tumors may evoke resistance in treatment through a shift in antigen presentation, other activating immune checkpoints, or the recruitment of immunosuppressive cells. And in autoimmune diseases the immune system can evade targeted therapies by finding alternate inflammatory pathways or by generating anti-drug antibodies.

Another problem is whether it is possible to achieve lasting immune responses and remission. In cancer, immune cell fatigue and immunosuppressive tumor conditions confine the development of memory T cells required to achieve long-term protection. In autoimmune diseases, immune tolerance is usually not achieved, and the disease comes back again after discontinuing the therapy. The solutions to these problems, therefore, need to improve the target delivery, tissue penetration, and immune memory [21]. Nanomedicine platforms of delivery provide some potential solutions that allow delivering immunomodulators with specificity, increased intratumoral drug delivery, and control of the release, which form the basis of an improved immunotherapy response and reduced systemic toxicity.

## 4. Nanomedicines: A Paradigm Shift in Immunotherapy

Nanomedicine has now opened up as a paradigm-shifting approach to the field of immunotherapy, providing the unprecedented promises of regulating the immune responses with improved precision, efficacy, and safety. As compared to typical immunotherapeutic methods, which can be ridden with undesired systemic toxicity, dismal bioavailability, and limited immunity-targeting capabilities, nanoscale therapeutics have been found to be highly specific multi-faceted modulators of the immune system based on exclusive physiochemical facilities, that is, size, form, surface charge, and versatile functionality. With their better pharmacokinetics and patterns of biodistribution, such nanomaterials like lipid-based nanoparticles and polymeric carriers, and inorganic systems such as gold and silica, have been given the ability to preferentially deliver immunotherapeutic agents to immune cells and tumor tissues. They have the advantage of their ability to penetrate biological barriers, e.g., tumor microenvironment (TME) and lymphatics, and even the blood–brain barrier (BBB), to treat diseases which cannot be treated with standard methods [22].

The above aspect is one of the major strengths of nanomedicine, which is its ability to combine several functionalities on the same platform. The double-purpose nanoplatforms are not just constructed to deliver therapeutic payloads, but they also have the capability to facilitate real-time imaging that allows them to specifically target a pathological location. This theranostic possibility enables diagnosis and treatment, plus immune monitoring, all at the same time, to give dynamic feedback of efficacy of therapeutics. Selectivity in targeting immune cells or tumor-selective markers is achieved using active targeting mechanisms in which molecules are attached to the surface (ligands such as antibodies, peptides, or aptamers) [23]. Simultaneously, the passive targeting is advantaged by the enhanced permeability and retention (EPR)effect, namely in solid tumors. These systems are also designed to deliver different immunomodulators, such as checkpoint inhibitors, cytokines, nucleic acids, and tumor-associated antigens, further boosting antigen presentation, dendritic cell activation, and a response of cytotoxic T lymphocytes, and reducing the toxicity of collateral damage. As an example, lipid nanoparticles (LNP) systems that have been strongly confirmed in mRNA vaccine technologies in the context of the COVID-19 pandemic are being actively investigated in cancer and autoimmune immunotherapies to conclude with high accuracy in immune modulation [24].

Another important milestone enabled by nanomedicine is the greater precision of delivering treatment to immune cells or at a tumor site. For cell-specific delivery, nanoparticles can be targeted via receptor-targeting endocytosis, the stimuli’s favorable pH, redox, or enzyme activity in diseases tissue. This specific translation not only enhances the therapeutic index but also reduces the collateral tissue damage to the healthy tissues which is vital when treating autoimmune diseases. Further targeting specificity and biocompatibility is also obtained through bio-mimetic systems such as exosome-mimetic vesicles and cell membrane-coated nanoparticles, providing novel immune editing strategies with technology such as CRISPR/Cas9. More recently, there are strategies including the use of hybrid lipid-polymer nanoparticles that display high specificity towards tumor-infiltrating lymphocytes (TILs) that allow isolated immune activation with maintained systemic homeostasis, an important aspect of autoimmune diseases [25].

Notably, nanomedicine is becoming a part of a wider immune-oncology plans and autoimmune tolerance modalities. Oncology nanoscale platforms potentiate traditional modalities of immune checkpoint inhibitors, CAR-T cell therapy, and cancer vaccines. To illustrate, providing a co-delivery of nanoparticles using anti-PD-1 and anti-CTLA-4 antibodies alongside tumor antigens, one can attempt to organize a multifaceted blockade of pathways of immune evasion and at the same time prime T-cells with binding antigen responses. The combinatorial strategies are actively studied as preclinical and clinical solutions to treat problematic malignancies. On the other hand, in managing autoimmune disease, nanotechnology is used in inducing antigen-specific tolerance through the selective delivery of autoantigens in the context of tolerogenic therapies to expand regulatory T-cells and dampen autoreactive immune responses. An example of the use of this approach is biodegradable PLGA nanoparticles loaded with myelin oligodendrocyte glycoprotein (MOG) peptides, which has shown efficacy in models of multiple sclerosis mediated by immune tolerance. Moreover, potential applications of nanomedicine in cancer and autoimmunity are further extended by use of artificial antigen-presenting cells (APCs), which are assembled on nanoscale scaffolds with co-stimulatory molecules and MHC-peptide complexes, allowing a precise control of T-cell repertoires [26]. Thus, nanomedicine is a paradigm shift in the field of immunotherapy through allowing multifunctional and specific but targeted modulation of immune response. Nanotechnology is at the stage of redefining the therapeutic effects of cancer and autoimmune diseases through its interrelated applications with new immune-oncology interventions and tolerance-inducing platforms. Recent breakthroughs made in materials science, in combination with translational and clinical confirmations, will further define the future landscape of immunotherapeutics driven by nanomedicine.

## 5. The Relationship Between Immunotherapy and Nanomaterials

### 5.1. Mechanism of Action in Cancer vs. Autoimmune Immunotherapy

Immunotherapy is based on re-setting the immune activity, but the course of therapy of cancer and autoimmune conditions takes opposite directions. The immune system in malignancies is usually functionally suppressed, allowing the malignant cells to grow despite them being immunogenic. The immunotherapy of cancer is thus directed to rejuvenate ineffective cytotoxic and antigen-reactive immunological pathways. Its key pathway is the successful restoration of effective awareness of abnormal cells through a heightened co-stimulatory signaling, the establishment of T-cell responsiveness, and antigen-driven proliferation of effector lymphocytes [27]. Nanomedicine reinforces these plans in that the immune-stimulating cues, be it molecular triggers or encoded antigens, arrive at immune structures where activation should be achieved, especially lymphoid interfaces. Nanosystems using advanced physicochemical design can enhance antigen processing, encouraging beneficial intracellular processing and sustained antigen immunology to facilitate powerful antitumor priming. During autoimmunity, the body does not suppress the immune effect but rather produces an excessive or misplaced immune response to the self-parts. Consequently, the development of therapeutic tools should minimize the unsuitable activation and not discard global immunity. Autoimmune immunotherapy, as a mechanism, aims at correcting antigen interpretation and regaining regulation supremacy. Nanomedicine also plays its part by offering immune-modifying cues as non-inflammatory forms and so directing immune cells toward non-inflammatory signaling programs. Nanosystems have the potential to change differentiation programs to regulatory phenotypes and mitigate persistence of pathogenic clones by modulating antigen display and antigen perception by intracellular pathways [28].

Therefore, disease contexts make use of identical immune recognitions modulation. Although cancer therapy enhances immune vigilance, autoimmune therapy restores restrained, self-protective immunity. The nanomedicine provides an interface that allows sculpting of immune behavior in both directions with considerably greater control than with traditional methods.

### 5.2. Current Immunotherapeutic Modalities

There is a growing array of clinically approved and under-study approaches for the treatment of cancer and autoimmune diseases. Such treatments are beginning to meet with nanotechnology to enhance effectiveness and minimize side effects.

Cancer Immunotherapy
Immune Checkpoint Inhibitors (ICIs): The PD-1, PD-L1, and CTLA-4 antagonistic antibodies which (by removing the inhibited immune pathways) enhance T-cell effector functions against cancer. The recent report on nanotechnology development suggested that it was possible to improve the pharmacokinetics and tumor-targeted delivery of ICIs, as well as improve the therapeutic effect and decline the systemic exposure. Despite the evidence presented in preclinical research, the possibility of nanoparticle-mediated delivery being used to decrease the burden of immune-related adverse events (irAEs) needs to be reconfirmed by being studied in clinical trials [29].CAR-T Cell Therapy: CAR-T cell therapy entails the genetic dressing of patient-derived T cells to express chimeric antigen receptors (such as CD19 CAR-T against B-cell malignancies). Although the present manufacturing protocol is complicated, requiring an ex vivo viral transduction, researchers are continuing working on the nanomaterials-based platforms such as lipid nanoparticle, polymers, hydrogels, and implantable scaffolds as a potential system to modify T cells in ex vivo (e.g., non-viral by intravenously delivering CAR-encoding nucleic acids to circulating T cells [30]).Cancer Vaccines: Cancer vaccines which are designed on personalized neoantigens which are specific to tumors, especially when made with a nanoparticle platform that allows high uptake in antigen-presenting cells and provides a better cross presentation. Preclinical data indicate robust activation of CD8^+^ T-cells, and the data in the clinic suggest efficacy, e.g., mRNA-based lipid nanoparticle vaccines against melanoma and pancreatic or kidney cancer. Nevertheless, although the nanoparticle delivery approaches may be promising, clinical validation of nanoparticle based neoantigen vaccines other than the lipid mRNA constructs is limited and underway [31].Oncolytic Viruses and Cytokine Therapies: Engineered oncolytic viruses that have been engineered to secrete cytokines (e.g., IL-2, IL-12, GM-CSF, IL-15) are often coupled with nanocarriers like nanovesicles (extrinsic coats or cell based carrier) to allow targeted co-delivery with adjuvants, minimize clearance by neutralizing antibodies, and activate immune cells within the tumor microenvironment [32].Autoimmune Immunotherapy
Cytokine Modulation: The examples of biologic therapy are anti-TNF alpha (e.g., infliximab and adalimumab) and IL-6 inhibitors (e.g., tocilizumab), which act to cut down the systemic inflammatory cascade by selectively neutralizing key cytokines implicated in the pro-inflammatory signaling. These treatments are also clinically proven and effective in cytokine storm syndrome treatment and in autoimmune and inflammatory disease Treatment [21] (Valencia et al., 2019).B Cell Depletion: Anti-CD20 monoclonal antibodies such as rituximab destroy selectively CD20^+^ B cells, and they are employed off-label in autoimmune diseases, such as Systemic Lupus Erythematosus (SLE) and multiple sclerosis (MS). Rituximab is also not FDA approved with such indications but has exhibited clinical effectiveness in specific patient categories with refractory SLE and related to relapsing multiple sclerosis, and effects have been principally obtained through the immunomodulatory and cytotoxic effect the drug has had on the B cell populations itself [33].Antigen-Specific Tolerance Induction: Autoantigens can be presented in a tolerogenic environment by the use of nanoparticles (often to dendritic Cells and including delivery of immunomodulatory agents) to induce regulatory T cells and lead to antigen-specific tolerance instead of immune activation.Nanoparticle-Based Tolerance Therapies: Tolerance therapies using nanoparticles-based approaches are in active preclinical development with type 1 diabetics, multiple sclerosis, and, to a lesser degree, rheumatoid arthritis. The majority of them are disease-specific self-antigens delivered via biodegradable polymer carrier, typically poly-lactic-co-glycolic acid (PLGA) and poly-lactic acid (PLA) in efforts to re-educate antigen-presenting cells, drive T cell anergy or apoptosis and increase antigen-specific regulatory T cells with or without addition of tolerogenic agents (e.g., dexamethasone or rapamycin). It has been found to be efficacious in animal models in EAE and NOD mice. Although early clinical trials (e.g., TIMP platforms in celiac disease and MS) are continuing, no composition has so far obtained regulatory approval. The putative field is still plagued with the single antigenic heterogenicity, dose scheduling, and an ideal nanoparticle design associated with immune tolerance with persistence and specificity [34].

### 5.3. Classification and Types of Nanomedicine 

Nanomedicines are synthetic nanomedical systems (1–1000 nm) intended to deliver unique drug delivery, slow release, and increased bioavailability of medicinal products. They connect material science and biomedicine to remove the classical pharmacological boundaries, especially in immunotherapy. Nanomedicines are stratified depending on their physiochemical properties and composition, in the following manners (Table 1):

#### 5.3.1. Liposomes and Lipid-Based Nanocarriers

Liposomes are vesicles of phospholipids bilayers of the spherical shape, which can encapsulate not only hydrophilic drugs but also hydrophobic ones. They are interesting in immunotherapy due to their biocompatibility, adjustable surface properties, and plasticity with cell membranes. In cancer, liposomal doxorubicin (Doxil^®^) minimizes cardiotoxicity and enhances tumor penetration through the enhanced permeability and retention (EPR) effect [3]. Newer technologies involve immunoliposomes whose surface is conjugated with antibodies or ligands that specifically target immune cells or microenvironments of tumors actively. In autoimmune diseases, liposomes have been employed to give direct delivery of immunosuppressants (e.g., methotrexate or dexamethasone) to the inflamed tissue and thereby minimize systemic toxicity. Moreover, the development of tolerogenic liposomes peptides of the self with immunomodulators to obtain antigen-specific immune tolerance is underway.

#### 5.3.2. Polymer-Based Nanoparticles

Nanoparticles of polymers, which are polymers that are biodegraded, e.g., poly(lactic-co-glycolic) acid (PLGA), poly(ethylene) glycol (PEG), or chitosan, provide the advantage of using tailored drug loading and release powers. They can have their version altered so as to have active targeting or stealth quality. They have been used in oncology, as the PLGA nanoparticles from encapsulation of siRNA or chemotherapeutics showed increased tumor targeting and decreased off-target effects. Remarkably, polymeric carrier-based cancer nanovaccines have been reported to provide encouraging results in the dendritic cell activation and cytotoxic T cell responses [35]. PLGA nanoparticles are being tested to deliver autoantigens and immunomodulators to the lymphoid organs in the treatment of autoimmunity to promote regulatory T cell responses. A potential platform is the use of antigen-loaded nanoparticles combined with rapamycin and used to induce tolerance in experimental autoimmune encephalomyelitis (EAE) models.

#### 5.3.3. Inorganic Nanomaterials

The inorganic nanomaterials such as gold nanoparticles (AuNPs), silica nanoparticles, quantum dots, and iron oxide nanoparticles have particular optical, magnetic, as well as catalytic properties. They are functionalizable on their surfaces to bio-interactions and so can be applied in therapeutic or diagnostic (theranostic) applications. The use of gold nanoparticles as carrier in combination with tumor antigen and adjuvant to boost the Potency of cancer vaccines has been documented. Since iron oxide nanoparticles have a magnetic property, they have been explored in the targeted drug delivery systems as well as the tracking of immune cells in vivo [36]. In autoimmunity, nanoparticles of silica have been tested to scavenge pro0inflammatory cytokines or small-molecule inhibitors to specific pro-inflammatory sites to show a reduction in systemic immunosuppression.

#### 5.3.4. Biomimetic and Hybrid Nanoplatforms

It is anticipated that biomimetic nanoparticles look like biological scaffolds, e.g., exosomes, cell membrane, or viral capsules, to acquire the immune compatibility and targeting. Multifunctional hybrid nanoplatforms use organic and inorganic materials. In the case of cancer immunotherapy, nanoparticles coated with tumor cell membranes may achieve tumor antigen presentation but avoid immune clearance to facilitate an effective tumor antigen presentation and T cell priming. There has also been potential in the ability of exosomes-mimetic nanovesicles to deliver RNA-based immunomodulators. In autoimmune disorders, through engineering, nanoparticles usage is carried out to be coated by leukocyte membranes and used as decoys, which bind and neutralize the inflammatory cytokines or pathogenic autoantibodies [37]. Such smart systems minimize damaging tissue collateral and do not damage systemic immunity.

### 5.4. Nanomaterial-Immune Interactions: Safety, Biocompatibility, and Immunogenicity

Immune system, combined with engineered nanomaterial, plays a key role in the success of nanomedicine in translation to both oncologic and autoimmune immunotherapies. As more and more nanosystems are being developed with the ability to modify the immune system, whether they are the carriers of an immunotherapeutic drug or themselves act as immune-stimulators, their immune profile and their immunological footprint will need to be explicitly defined.

Biocompatibility denotes that the nanomaterial is congruent with carrying out its necessary action without evoking harmful local or systemic effects. Key parameters are particle size (less than 100 nm in size are drained by the lymphatic system), topology of the surface, zeta potential, as well as chemical composition. to avoid opsonization, activation of the complement system, and clearance by the mononuclear phagocyte system (MPS) prematurely, methods to avoid possible triggers include leukocyte-derived vesicles used as cloaking to avoid opsonization and subsequent effects of the complement system [38]. Nonetheless, recent evidence indicates that the repeated injection of the PEGylated nanoparticles can result in an anti-PEG immune response, which makes it difficult to work on a long-term basis.

Immunogenicity occurs in two forms: a negative hypersensitivity/cytokine release immune activation or beneficial immunostimulant treatment response. Take as an example those nanoparticles with encapsulated toll-like receptors (TLR) ligands (e.g., CpG-ODN, poly I:C) and neoantigens which, as a powerful dendritic cells maturation and cross-presentation via MHC-1 stimulator, trigger active CD8^+^ cytotoxic T lymphocyte (CTL) activation which is the key to antitumor immunity. Faulty innate immune sensing as mediated by the NLRP3 inflammasome, or the STING pathway, however, may well be connected with systemic inflammation or even autoimmunity [39].

The systemic delivery of nanoparticles causes the dramatic change in the immunological identity caused by protein corona deposition. Immunogenicity may come in the form of dynamic adsorption of plasma proteins (e.g., immunoglobulins, complement factors, apolipoproteins) and turn stealth nanoparticles into immunogenic objects, which have different cellular uptake, cellular trafficking, and cytokine secretion profiles.

## 6. Nanomedicine in Cancer Immunotherapy

Cancer immunotherapy has become a revolutionary practice in the treatment of cancerous diseases, which now aspires to reinstate and elevate the powers of immune system to identify and eliminate cancerous cells. Even though it has succeeded, its broad efficacy is still limited by several hurdles that include immunosuppressive tumor microenvironment (TME), ineffective antigen presentation, intra-tumoral migration of immune cells, and immune-related adverse effects. Nanomedicine is an extremely valuable system to address these challenges as it would allow precise, multifunctional, localized delivery of immunomodulatory agents. Combining nanoscale engineering with immunology, new ways have been established to remodel the immune landscape, improve antigen-specific response, and enhancing efficiency.

### 6.1. Modulating the Tumor Immune Microenvironment and Targeting Immunosuppressive Cells

Tumor microenvironment (TME) is an active, immunosuppressive niche where both the expression pattern of anti-inflammatory cytokines (e.g., IL-10, TGF-beta) as well as the suppressor cell populations (In other words, the regulatory T cells [CD4^+^CD25^+^FOXP3^+^], myeloid derived suppressor cells [CD11b^+^Gr1^+^], and M2-polarized tumor associated macrophages [CD206^+^, IL10^+^]) is hyperactive. The Immunosuppressive signaling pathways such as STAT3, TGF-receptor I/II and IDO1 have been targeted by small molecule inhibitor, siRNA, or CRISPR/Cas 9 loaded on polymeric and inorganic particles. For example, polymeric nanocarriers that contain TGF-beta receptor kinase inhibitors interfere with SMAD 2/3 phosphorylation, interfering with Treg differentiation and activation of fibroblasts. Iron oxide nanoparticles mediated oxidative stress/induced lysosomal disruption by ROS, which resulted in immunogenic macrophage polarization through activation of the NF-kB pathways and promoted an M1-like phenotype supported by elevated levels of IL-12 and TNF-α secretion [40].

Specific ablation of MDSCs can be performed by gold nanoparticles functionalized with chemokine receptor 2 (CCR2) antagonists or CXCR4 inhibitors, which can inhibit the chemokine-driven transport of suppressive myeloid cells to tumor beds. Furthermore, nitric oxide-donor-releasing-hypoxia-responsive nanoparticles regulate the HIF-1 signaling by promoting maturation of dendritic cell (DC), expression of MHC-I/II, and establishing infiltration of T-cells. By adding functionabilities to circumvent the endosome, e.g., with proton-sponge polymers, or fusogenic peptides, the targeted siRNAs can then deliver to the cytoplasm and block the expression of the transcripts of the FOXP3 or CCL5 with corresponding effects in Tregs, reinvigorating effector-exhausted CD8^+^ cells [41]. Therefore, nanomedicine remodels the immune space of solid tumors and provides lasting anti-tumor immunity in a mechanistically driven manner.

### 6.2. Nanocarriers for Cancer Vaccines, Neoantigen Delivery, and Immune Checkpoint Modulation

Success of immunotherapeutic approaches is focused on antigen-specific immunity. Nevertheless, one can degrade the immunogenicity of tumor-associated antigens (TAAs) and neoantigens because of the inefficient dendritic cells priming, inefficiency in presenting antigens, and destruction in the endolysosomal compartments. These shortcomings can be overcome by nanoparticles-based platforms of cancer vaccines based on the rational design of antigen-adjuvant co-delivery systems with intended physiochemical characteristics and immunological properties.

mRNA that encodes personalized neoantigens epitopes has been delivered using lipid nanoparticles (LNPs) that have also been optimized to improve endosomal escape and trafficking to lymph nodes. Such LNPs can be loaded with ionizable lipids and cholesterol/PEG-lipids and helper lipids to enable mRNA condensation, membrane fusion, and cytosolic processing, leading to the neoantigen expression with high fidelity in CD11c^+^ DCs. Introduction of MHC-1-restrained epitopes results in vigorous stimulation of CD8^+^ effector T cells and formation of T cell conservation pools [42]. Polypeptide-based scaffolds or artificially produced peptide amphiphiles enable constructing self-assembling nanovaccines that can accurately control multivalency and spacing between antigens, just as viral capsids do, which allows better targeting of a vaccine. Type 1 interferon can be induced using Toll-like receptor (TLR) ligand incorporation, including CpG-ODN (TLR9) or poly I:C (TLR3), to further increase the co-stimulatory molecules (CD80/CD86) expression on the APCs [43].

Under the immune checkpoints perturbation setting, nanocarriers enable the restriction of anti-CTLA-4 monoclonal antibodies. pH-responsive micelles and redox-responsive vesicles utilize the acidity and reductancy in the tumor microenvironment to deliver ICIs specifically, allowing reductions in systemic immune related adverse events (irAEs). Nanoparticle-based TASCO (T Cell Activation by Simultaneous Costimulation) and Checkpoint Blockade drugs take advantage of the enhanced costimulatory and T cell receptor (TCR) signaling simultaneously by co-encapsulating co-stimulatory ligands (e.g., agonistic anti-OX40 or CD137 ligands) and checkpoint inhibitors (e.g., anti-PD-1) within dual delivery nanoparticles [44]. This results in a potent synergistic activation of T cell costimulatory and T cell receptor signal transduction that can both potentiate the cytot.

New nanoformulations such as CRISPR/Cas9 loaded lipid-polymer hybrid nanoparticles can also break the expression of PD-L1 in tumor cells by targeted reprogramming of the genome and provide more significant blocking of checkpoints over time. siRNA-mesoporous silica nanoparticles targeting LAG-3, TIM-3, or VISTA extend the tools available to block the checkpoint, especially in PD-1 axis-resistant tumors [45]. All these approaches of nanocarriers redefine the administration and efficacy of cancer vaccines and checkpoint therapeutics with enhanced antigen persistence, minimal toxicity, and powerful activation of adaptive immune responses.

### 6.3. Enhancing Adoptive Cell Therapies and Combinatorial Immuno-Nano Strategies

The unprecedented success of adoptive cell therapies (ACTs), including CAR-T and TCR-T cell therapies in hematological malignancies, has no parallel in solid tumors, which are still limited by antigen heterogeneity, physical barriers in the TME, and T cell dysfunction. Nanotechnology plays a role in promoting the pharmacodynamic environment of ACTs by advancing tumor-homing, durability, and activation of T cells in the tumor locality. Cytokine-release nanoscaffolds that maintain localized levels of IL-15, IL-21, or Il-2 in the peritumoral niche, which enables T cells to expand and avoid terminal exhaustion, have been designed using biodegradable scaffold materials of alginate, PLGA- or PEG-based hydrogels. Nanocarriers expressing surface-coupled chemokines (e.g., CXCL9/10) or adhesion ligands (e.g., VCAM-1 mimetics) enable specific targeting and trapping of CAR-T cells as tumor sites [46].

A new frontier is in situ T cell engineering through nanoparticle-mediated gene transfer. Nanoparticles bearing CAR-encoding mRNA or transposons may transfect circulating or tissue-dwelling T cells to render functional CAR without the need of ex vivo manipulation. The latter formulations frequently use electroporation-independent cytosolic delivery vehicles (cell-penetrating peptides, ionizable lipids, or microfluidic shear-based fragmentation) to dramatically lower cost and manufacturing lead time as well as enhance access to ACT.

The synergy between ACTs and immunogenic cell death (ICD)-inducing chemotherapeutics, radiosensitizers, or immune adjuvants has been attempted combining combinatorial immune-nano approaches. Oxaliplatin and anti-CD47 antibodies nanoparticle are co-loaded to co-induce calreticulin exposure, ATP liberation, and translocation of HMGB1, which by itself leads to phagocytosis and antigen cross-presentation. In combination with CAR-t or TCR-T treatment, this method leads to an increased clonal expansion and a better destruction of tumors [47].

Other techniques of enhancing ACT using the concept of photothermal and photodynamic nanomedicine include providing localized ablation of tumors, normalization of vessels, and releasing antigens. Photo-activatable nanoparticles with IR dyes or porphyrins induce localized oxidative stress causing release of neoepitopes to expand the T cell repertoire which decreases escape. These mechanistically enabled platforms are the future of immune-nanotechnology, and they have the possibility of immune cell fate control, antigen presentation, and immune activation in real time in the solid tumor environment.

## 7. Nanomedicine in Autoimmune Immunotherapy

Autoimmune processes are facilitated by the loss of self-tolerance, the inappropriate up-regulation of autoreactive T and B cells, increase in the synthesis of inflammatory mediators (pro-inflammatory cytokines), and the destruction of tissues, or systemic tissue destruction. Many conventional immunosuppressive drugs do not have antigen specificity and are very toxic, affecting most body systems with resultant immunodeficiency. The promise of nanomedicine, provided by the rationally engineered nanocarriers, is an innovative approach, which can provide antigen-based immune manipulation, targeted immune-regulatory agent delivery, and remodeling of dysfunctional immune networks. Nanoscale delivery systems-polymeric nanoparticles, liposomes, exosomes, dendrimers, carbon-based engines, and metal–organic frameworks enable spatiotemporal control of immunotherapeutic payloads much more than the conventional systems, with increased efficacies and minimized off-target effects.

### 7.1. Immune Tolerance Induction and Lymphoid Targeting via Nanomedicine

One of the most important objectives of autoimmune therapy is to restore immune tolerance but not to undermine overall immune performance. In nanomedicine, antigen-specific tolerance is achieved by designing transportation of self-antigens to tolerogenic antigen-presenting cells (APCs), especially of immature dendritic cells (iDCs) and marginal zone macrophages (MZMs), within the lymphoid tissues. Mannose, DEC-205, or DC-SIGN-conjugated nanoparticles achieve active specificity of Dc subsets in the secondary lymphoid organs. When taken into cells, NPs containing bio-degradable material (e.g., PLGA, PEGylated PLA) or made of lipids release their encapsulated auto antigens (e.g., MOG, insulin peptide, GAD65) in the endosomal pH which can present themselves in MHC Class II, enabling the auto antigen to bypass dangerous associated molecular patterns (DAMPs) [48]. It results in activation of co-inhibitory molecules (e.g., PD-L1, ILT3) and release of TGF-β that in turn promotes the de novo differentiation of FOXP3^+^ regulatory T cells (Tregs) and provokes the clonal anergy or deletion of effector memory T cells.

Also, nanoparticles can be preferentially gathered within lymph nodes (LNs) and splenic white pulp by passive (EPR effect) or active (surface ligand-conjugated) mechanisms, an activity that reduces the exposure of antigens to immune dystrophy in an enduring tolerogenic milieu. This strategy works quite well in cases involving multiple sclerosis, type 1 diabetes, and autoimmune uveitis as the diseases that need the silencing of the immune system, which is antigen specific.

### 7.2. Nanocarriers for Cytokine Modulation and Adaptive Immune Reprogramming

Autoimmune diseases are characterized by Th1/Th17-polarized environment and elevated cytokines of pro-inflammatory types (e.g., TNF-alpha, IL-6, IL-17A, IFN-gamma) advancing immune-mediated inflammation and immune-cell-mediated destruction. Reprogramming nanocarriers to immunocyte specificity triggers cell-type-specific delivery of cargo that may include anti-inflammatory cytokines (IL-10, TGF-8), MTOR inhibitors (e.g., rapamycin), Janus Kinase (JAK) inhibitors, or small interfering RNAs (siRNA) that are directed against STAT 3, NF-kB, or IL-6R functionalities [49]. Case in point, miRNA-146a mimics or IL-10 mRNA can be delivered into macrophages by cationic lipid nanoparticles to provide M2 anti-inflammatory phenotype. In the same manner, administration of siRNA-STAT1 to monocytes/macrophages using dendrimeric NPs as a delivery tool inhibits the IFN-mediated inflammation that occurs during systemic lupus erythematosus (SLE). Mesoporous silica nanoparticles (MSNs) and metal–organic frameworks (MOFs) which allow co-delivery or sequential delivery of autoantigens with immune-suppressive agents provide amplification of synergistic immune modulation [50].

Autoreactive B cells and plasma cells also undergo reprogramming based on nanoparticles. Using B cell receptor (BCR)-targeted liposomes, functionalized with CD22 or BAFF-R ligands, allows the targeted delivery of tolerogenic agents or apoptotic signals, killing pathogenic B cell clones, which produce the autoantibodies that cause the disease, including SLE, RA, and Myasthenia Gravis [51]. On the T cell, synthetic antigen-presenting nanoparticles (APNs) conjugated with MHC-peptide complexes and co-inhibitory ligands (PD-L1, CTLA-4Ig) result in decoupling the signal 1 and 2, thus resulting in T cell exhaustion or deletion. Such a mechanism has proved to reverse T cell-mediated demyelination and cytotoxicity to β-cells.

### 7.3. Disease-Specific Applications and Translational Nanomedicine Advances

Rheumatoid Arthritis (RA):The pathology of RA includes hyperplasia of synovial, neovascularization, and infiltration by Th17 cells, macrophages, and fibroblast-like synoviocytes (FLS). The passively targeted delivery of nanoparticles loaded with methotrexate, TNF-α inhibitors, or JAK inhibitors to the inflamed synovium significantly relies on elevated permeability and retention (EPR) effect of the inflamed endothelium. pH-sensitive liposomes and thermosensitive hydrogels have been applied intra-articularly and show localized inhibition of the IL-6/STAT3 axis and preservation of joints (Table 2) [52].Multiple Sclerosis (MS):Myelin-specific tolerance has been established through lipid nanoparticles (LNPs) that transport MOG or PLP peptides together with IL-10 or TGF-β. Also, CD11c^+^ DCs-targeting nanogels PEGylated nanoparticle therapy regulates T cell in the CNS and decrease neuroinflammation without stimulating systemic immunosuppression [53].Type 1 Diabetes (T1D):Encapsulated nanocarriers containing insulin B-chain peptides and drug candidates like rapamycin or tolerogenic adjuvants, like vitamin D3 or dexamethasone, have been successful in renormalizing Treg/Teff ratios and avoiding destruction of the pancreatic β-cells. Targeting Langerhans cell function through nanoparticles in the subcutaneous area is effective in inducing mucosal tolerance, which can also be used as one of the strategies of intervention at an early stage [54].Systemic Lupus Erythematosus (SLE):TLR7/9-inhibitory-engineered nanoparticles focusing on plasmacytoid dendritic cells (pDCs) and B cells have been revealed to diminish type 1 IFN signaling, the development of the immune complex, and nephritis. A hydroxychloroquine and siRNA-TLR9 co-delivery system based on albumin-bound NP/Lipid-polymer hybrid are in preclinical development [55].Psoriasis and Inflammatory Bowl Disease (IBD):Topical or systematic administration of polymeric micelles or solid lipid nanoparticles loaded with IL-17 inhibitors, siRNA-STAT3, or TNF blockers have shown specific effects on IL-17 pathology at inflamed epithelial barriers [56].

## 8. Emerging Advantages and Transformative Breakthroughs in Nanomedicine

### 8.1. Controlled Release, Targeted Delivery, and Pharmacokinetic Optimization

Cancer and autoimmune therapeutics often have poor clinical efficacies due to off-target cytotoxicity, less than ideal pharmacokinetics (PK), and high rates of systemic clearance. A multifunctional solution is achieved through delivery platforms based on nanomedicines, as it allows access to direct control of release, site targeting, and pharmacokinetic adjustment of therapeutic payloads. A multiplicity of nanocarrier systems have been developed such as lipid (liposomes), solid lipid nanoparticles (SLNs), polymeric nanoparticle (PLGA, chitosan), nanogels, dendrimers, carbon-based carrier (graphene Oxide, carbon nanotubes), and metal/inorganic nanostructure (gold, silica, iron oxide) to boost the physiochemical stability, solubility, and bioavailability of hydrophobic and biologically labile agents.

Drug delivery by stimuli-responsive systems forms a breakthrough in drug delivery systems. Such nanocarriers are designed to react under the influence of endogenous triggers, including acidic pH, raised levels of glutathione (GSH), ROS concentration, matrix metalloproteinase (MMPs), and cathepsin, which increase in the tumor microenvironment (TME) or in autoimmune lesions. Alternatively, the spatiotemporal drug release can be controlled by external stimuli: ultrasound, light (NIR/UV), magnetic field, or heat gradient. Active targeting is performed by conjugating surface ligands on the target, e.g., monoclonal antibodies (anti-CD44, anti-EGFR), peptides (RGD, TAT), aptamers (AS1411), and small molecules (folic acid, mannose), to specifically act on overexpressed receptors on diseased cells [57]. The proposed ligand-receptor mediated endocytosis would greatly increase internalization of a cell, avoid multidrug resistance (MDR) transporters and also decreasing off-target deposition in non-pathological tissues.

By PEGylation, zwitterion surface modification and coating with hydrophilic polymers, pharmacokinetic characteristics are optimized, e.g., allowing a long systemic circulation half-life, reduced renal clearance, and escape of immune recognition, by the mononuclear phagocyte system (MPS). Such nanocarriers are associated with enhanced area under the curve (AUC), decreased clearance (CL) with regulated maximum concentration (C_max_), and a long terminal half-life leading to enhanced therapeutic deposits at the pathological regions and limiting systemic distribution [58]. Such advanced systems also allow co-delivery of two (or more) therapeutic agents, e.g., co-delivery of a chemotherapeutic with a checkpoint inhibitor agent, or a tolerogenic antigen with an immunosuppressive cytokine, in one nanoplatform. This enables the synergistic regulation of the innate and adaptive immune elements in a kinetically tunable manner, ushering a new beginning in multi-modal immune-nanotherapy.

The proof of the feasibility of nanomedicine-based immunotherapies in cancer and autoimmune disease has already been shown through numerous experimental and clinical studies that have examined the feasibility of these translational applications of nanomedicine. A number of nanoparticle systems have been developed in oncology, including liposomal doxorubicin, albumin-bound paclitaxel, and polymeric nanocarriers expressing immune stimulators, which have moved to clinical practice and trials have shown reduced systemic toxicity, improved drug accumulation, and increased antitumor immune activation [59]. Likewise, advances have been made in nanoparticle-enabled vaccine preparations, enhancing the antigen presentation and T-cell priming with some in early-phase clinical testing. Preclinical research in the case of autoimmune diseases has demonstrated the promising effects of tolerogenic nanoparticles loading autoantigens of a disease or immunomodulatory molecules in restoring immune homeostasis and suppressing pathological immune responses. Some of these have been developed to antigen-coupled biodegradable nanoparticles to treat multiple sclerosis and peptide-loaded nanocarriers to treat rheumatoid arthritis and have progressed to human trials with promising safety and immunological results. All these experimental and clinical results allow one to conclude that nanotechnology-based immunomodulation is not only a theoretical proposal but a treatment approach with active biological action and increasing translational evidence.

### 8.2. Reducing Off-Target Effects and Redefining Therapeutic Windows via Nanoscale Engineering

The unparalleled abilities of nanomedicine can optimize the therapeutic index (TI)-the ratio of toxic to therapeutic dose by reducing off-target toxicities and shifting drug effect towards the intended site of actions. The limitations of traditional systemic therapies are usually characterized by non-selective biodistribution leading to toxicological stress of non-affected organs, including bone marrow, liver, heart, kidneys, and gastrointestinal epithelium. These nanocarriers that have adjustable physiochemical characteristics, including size (30–150 nm), shape (spherical, rod, disc and filamentous), surface charge (neutral or slightly negative), and mechanical stiffness, are specifically designed to control extravasation, interstitial transport, and cellular uptake either by passive targeting (in principle based on the enhanced permeability and retention, or EPR effect) or active targeting [60]. As an example, within the context of autoimmune pathologies nanoparticles may be specially guided to pro-inflammatory microenvironments by developing responsiveness to pro-inflammatory cytokines (e.g., TNF-α, IL-6), hypoxia-inducible factors (HIF-1), or a changed redox potential in immune-infiltrated tissues. This can enable site-specific immunosuppression with conservation of systemic immune surveillance. Moreover, nanoparticle delivery vehicles towards the target architecture of the lymph nodes, mucosal immunological regions, or secondary lymphoid tissues provide local vetoing of autoreactive T cells and B cells.

Nanostructures endowed with nuclear localization signals (NLS) and endosomal escape peptides make efficient delivery of cytosolic and nuclear-targeted biopharmaceuticals (e.g., siRNA, miRNA, antisense oligonucleotides, mRNA-based therapeutics) and gene editing components (e.g., CRISPR/Cas9 RNPs) possible with minimal to no toxicity [61]. This type of nanosystem allows increased intracellular bioavailability and avoids enzymatic degradation in endolysosomal compartments, allowing effective action at lowest doses and significantly lowering the risk of toxicity.

The appearance of nanotheranostics, that is, the hybrid nanoplatforms combining diagnostic imaging and therapeutic capabilities, allows real-time observation of drug release, biodistribution, response to treatment using PET and MRI, photoacoustic imaging, or fluorescence resonance energy transfer (FRET) [62]. Such platforms enable the dynamic adjustment of the dosing schedules to patient-specific pharmacodynamics, resulting in the patient-specific dosing regimens and precision medicine approaches with the most significant therapeutic effects. Notably, the design of next-generation nanocarriers that can adjust to the biologically changing parameters during treatment is now being achieved through mathematical pharmacokinetic/pharmacodynamic (PK/PD) modeling, machine learning-based optimization, and integration into the systems biology. It is a change in paradigm which is moving toward predictive, personalized, and precision nanomedicine, not where toxicity is simply minimized, but where it is being designed out of the therapeutic equation through intelligent and adaptive nanocarriers.

## 9. Emerging Technologies and Future Trends

The convergence of nanomedicine and emerging biotechnologies is very quickly transforming the prospect of medicine to deal with cancer and autoimmune immunotherapy treatment. The emergent frontier is the design of very versatile, smart, and multifunctional nanosystems that are able to tailor immunomodulation, to deliver drugs precisely, and to follow therapy and monitor its effect in real-time. A significant development towards the latter is the potential utility of nanomaterials capable of responding to certain physiological conditions; these are bioresponsive nanomaterials designed to rather selectively release therapeutic loads upon encounter with a particular stimulus (e.g., pH concentrations, redox, enzyme action, or inflammatory cytokine concentration). These on-demand systems enable spatiotemporal control of the release at site of pathology, thus causing low systemic toxicity and high therapeutic efficacy. These vehicle platforms are being optimized to present co-administered antigens, adjuvants, or checkpoints inhibitors in temporal concert to high fidelity reprogramming of dysregulated immunity.

At the same time, machine learning (ML) and artificial intelligence (AI) algorithms are currently being used to formulate nanoparticle preparations and optimize them depending on patient-specific immunological and genomic fingerprints. Predictive modeling can be used to guide rational choice of the nanoparticles surface ligands, choice of combinations of payloads, and release kinetics depending on the disease phenotype [63]. The AI-based method is not faster in terms of preclinical-clinical translation of nanomedicine platforms but can also promote adaptive regimens of immunotherapy that adjust according to tumor sequences or immune flare-ups. The use of AI is also transforming high-throughput nanostructure screening to bio-distribution profiles toxicity data, and other pharmacokinetics in silico, thereby limiting the need to test most samples as in vivo experiments. Combination of CRISPR-Cas genome editor tools with nanotechnology is the next disruptive strategy in targeted gene-modulated therapy-immune manipulation. The specific editing of immunoregulation genes in T cells, antigen-presenting cells, or stromal cells in the tumor or autoimmune microenvironment with the CRISPR/nanoparticle hybrids are currently used. These split systems allow a localized editing of genes with deminimus off-target effects and immunogenicity. As an example, altering the immune escape mechanisms in cancer by silencing pro-inflammatory genes in macrophages or reprogramming the tumor-infiltrating lymphocytes is being done by deploying non-viral lipid or polymeric nanoparticles encapsulating CRISPR-Cas complexes [61].

Moreover, exosome-mimetic nanoplatforms and cell-derived vesicles are being explored as bioactive, biological compatible delivery platforms. Such vesicles resemble endogenous exosomes both in form and in their properties, allowing them to evade the immune system, circulate long term, and be delivered to sites of immune response or inflammation. Mimic exosomal engineered exosomes can transport miRNAs, siRNAs, cytokines, or peptides that regain immune tolerance or enhance anti-tumor immunity. The fact that they are of autologous origin minimizes immunogenic reactions, whereas the modular aspect allows customization of the cargo content to selective disease treatment.

At the same time, personalized and precision nanomedicine approaches are trending to immunophenotype-directed nanotherapy. The design of nanoparticles is being informed by patient stratification by biomarker expression, including PD-L1 expression, their T-cell infiltration profile, HLA haplotypes, or autoantibody signature [10]. These personal systems will mean that each patient is aligned to their particular immunological landscape, and the likelihood of failure will be low and of persistent response will be high. All this innovation is leading nanomedicine towards becoming a pillar of precision immunotherapy in the future with adaptive, patient-specific solutions to the complex immune disorders.

Nanotechnology continues to face a number of scientific, manufacturing, regulatory, and ethical constraints to the clinical translation of nanotechnology despite the great advances provided by the field. Another serious challenge is the inert predictable nano-bio interface: once in contact with biological fluids, nanoparticles obtain dynamic protein corona, changing their identity, their biodistribution, their immunogenicity, and their therapeutic efficacy, and preclinical predictions are no longer valid. Moreover, the improved permeability and retention (EPR) effect is among the widely publicized mechanisms that prospectively lack consistency in tumors in humans relative to animal models, lowering the effectiveness of passively targeted nanosystems. Manufacturing problems: There are also manufacturing problems such as the variability of batches to batches, non-straussability of the synthesis pathways, and a challenge in being able to exercise tight control over size, charge, and surface chemistry, which are barriers to scalable production, and international standardization. There are also toxicological uncertainties; nanoparticles can induce complement activation, oxidative stress, cytokine release, or long-term organ deposition, which makes one concerned with the chronic toxicity and immunogenicity. Nanomedicines do not have advanced regulatory pathways since the current regulatory frameworks cannot determine quality, safety, and effectiveness of multifunctional or hybrid nanosystems. Lastly, the expensive cost of development, the lack of accessibility, and the ethical issues of customized nano-therapy make it even harder to be used widely in the clinical practices.

## 10. Conclusions

The nanotechnology and immunotherapy combination have changed the paradigm of handling complex immune-mediated illnesses like cancer and autoimmunity. Nanomedicine can resolve inherent difficulties of immunotoxicity, therapeutic resistance, and non-specific systemic effects by providing control of delivery, nanoscale immune modulation, and improved pharmacokinetics and bioavailability. Multifunctional nanoplatforms, including responsive carriers and exosome-mimetic vesicles, CRISPR-engineered nanoparticles, and AI-designed delivery platforms, allow manipulation of immune elements, including antigen-presenting cells, T cell subunits, Cytokine constitutions, and immune checkpoints. The systems are used to mediate spatially restricted and temporally controlled modulation of immune responses, such as to achieve an enhanced antitumor responses or reconstitution of immune tolerance in autoimmunity. Also, nanomedicine enhances pharmacokinetic and pharmacodynamic signals, expands the therapeutic window, and facilitates combination signatures. Further developments of smart, adapting nanotherapeutic systems will be front and center in the development of a next-generation paradigm of safe, effective, personalized immune therapies across a wide range of diseases as research and development continues.

## Figures and Tables

**Figure 1 ijms-26-11941-f001:**
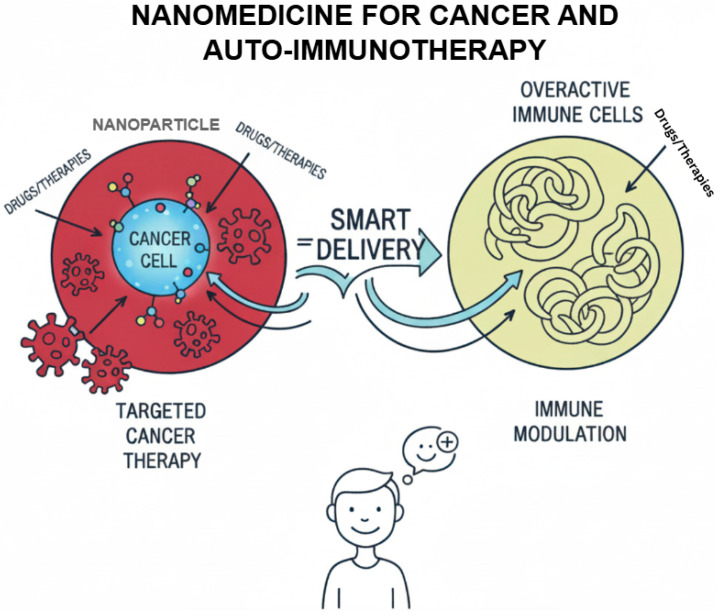
Nanomedicine for cancer and auto-immunotherapy.

**Figure 2 ijms-26-11941-f002:**
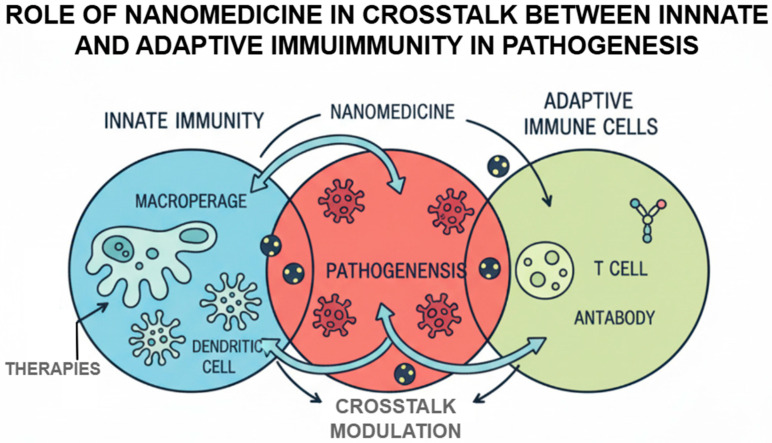
Role of nanomedicine in crosstalk between innate and adaptive immunity in pathogenesis.

**Table 1 ijms-26-11941-t001:** Classification of nanomedicine platforms.

Nanomedicine Class	Composition	Key Features	Representative Applications
Lipid-based nanoparticles (LNPs, liposomes)	Phospholipids, Cholesterol	Biocompatible, encapsulate hydrophilic/hydrophobic drugs	mRNA cancer vaccines, antigen delivery, siRNA PD-L1 silencing
Polymeric nanoparticles (PLGA, PEG, Chitosan)	Biodegradable polymers	Controlled release, tunable degradation	Tolerogenic nanoparticles for MS/T1D, cytokine delivery
Inorganic nanoparticles (gold, iron oxide, silica)	Metal Cores	Imaging + therapy, easy functionalization	Photothermal cancer therapy, macrophage repolarization
Biomimetic nanoparticles	Cell membranes, exosomes	High payload, multivalent surfaces	Tumor antigen-coated vesicles, exosome-mRNA immunotherapy
Dendrimers	Branched polymers		Gene delivery, TLR agonist delivery

**Table 2 ijms-26-11941-t002:** Disease-specific applications and translational nanomedicine advances.

Disease	Pathophysiology/Target	Nanomedicine Strategy/ Therapeutic Approach	Key Outcomes/Notes	Reference
Rheumatoid Arthritis (RA)	Synovial hyperplasia, neovascularization, infiltration by Th17 cells, macrophages, fibroblast-like synoviocytes (FLS)	Passively targeted nanoparticles (methotrexate, TNF-α inhibitors, JAK inhibitors); pH-sensitive liposomes; thermosensitive hydrogels	Localized inhibition of IL-6/STAT3 axis; joint preservation	Emami and Ansarypour, 2019 [52]
Multiple Sclerosis (MS)	CNS inflammation; myelin-specific autoimmunity	Lipid nanoparticles (LNPs) delivering MOG or PLP peptides + IL-10/TGF-β; PEGylated nanogels targeting CD11c^+^ DCs	Regulates CNS T cells; decreases neuroinflammation without systemic immunosuppression	Yuan et al., 2014 [53]
Type 1 Diabetes (T1D)	Autoimmune destruction of pancreatic β-cells; dysregulated Treg/Teff ratio	Encapsulated nanocarriers with insulin B-chain peptides + rapamycin/vitamin D3/dexamethasone; subcutaneous targeting of Langerhans cells	Renormalizes Treg/Teff ratios; induces mucosal tolerance; prevents β-cell destruction	Jung et al., 2023 [54]
Systemic Lupus Erythematosus (SLE)	Type 1 IFN signaling, immune complex formation, nephritis	TLR7/9-inhibitory nanoparticles targeting pDCs and B cells; co-delivery of hydroxychloroquine + siRNA-TLR9 Via albumin-bound NP/lipid-polymer hybrids	Diminishes IFN signaling; reduces immune complex development and nephritis	Satterthwaite, 2021 [55]
Psoriasis and Inflammatory Bowel Disease (IBD)	IL-17-driven epithelial inflammation	Polymeric micelles or solid lipid nanoparticles carrying IL-17 inhibitors, siRNA-STAT3, or TNF blockers	Specific targeting of IL-17 pathology at inflamed epithelial barriers	Yadav et al., 2021 [56]

## Data Availability

No new data were created or analyzed in this study. Data sharing is not applicable to this article.

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
