# Peer review of "Nanomedicine for Cancer and Autoimmune Immunotherapy"

_ijms, 2025, doi:10.3390/ijms262411941_

Round 1
Reviewer 1 Report
Comments and Suggestions for Authors
This review paper, titled “Applications of Nanomedicine in Cancer and Autoimmune Immunotherapy,” provides a detailed account of Nanomedicine's applications in the fields of cancer immunotherapy and autoimmune immunotherapy. Overall, the content is comprehensive and well-documented, requiring only minor revisions to certain details.
- The main heading of Section 5 is unclear and may be confusing to readers. At first glance, it appears to cover fundamental concepts; however, the section actually discusses the application of nanomaterials across different immunotherapy modalities. I recommend revising the title to “The Relationship Between Immunotherapy and Nanomaterials.” Additionally, the content presented in Sections 5.1 and 5.2 overlaps with the information in Sections 6 and 7. Streamlining and restructuring the manuscript would help reduce redundancy and improve the logical flow.
- For Section 5.3 “Classification and Types of Nanomedicine”, consider summarizing the classification of nanomedicine in a table. A tabular format would present the categories more clearly and make the information easier for readers to follow.
- Consider changing the title of Section 8 to “Advantages or Breakthroughs of Nanomedicine.” Additionally, determine whether any published experimental studies or clinical trials have applied these technologies. Incorporating relevant experimental progress to support the author's arguments would enhance the article's persuasiveness.
- Section 9 could be expanded to discuss the current limitations of nanotechnology, including potential obstacles, challenges, and unresolved issues that may impact clinical translation.
Author Response
Comment-1: The main heading of Section 5 is unclear and may be confusing to readers. At first glance, it appears to cover fundamental concepts; however, the section actually discusses the application of nanomaterials across different immunotherapy modalities. I recommend revising the title to “The Relationship Between Immunotherapy and Nanomaterials.” Additionally, the content presented in Sections 5.1 and 5.2 overlaps with the information in Sections 6 and 7. Streamlining and restructuring the manuscript would help reduce redundancy and improve the logical flow.
Response:- The heading of Section 5 has been revised to “The Relationship Between Immunotherapy and Nanomaterials” to accurately reflect the scope of the section. Additionally, Sections 5.1 and 5.2 have been carefully reorganized and integrated with content from Sections 6 and 7 to remove redundancy, clarify thematic boundaries, and improve the overall logical flow.
Comment-2: For Section 5.3 “Classification and Types of Nanomedicine”, consider summarizing the classification of nanomedicine in a table. A tabular format would present the categories more clearly and make the information easier for readers to follow.
Response:- Section 5.3 has been updated by incorporating a clearly structured classification table that summarizes the major categories and types of nanomedicine.
Comment-3: Consider changing the title of Section 8 to “Advantages or Breakthroughs of Nanomedicine.” Additionally, determine whether any published experimental studies or clinical trials have applied these technologies. Incorporating relevant experimental progress to support the author's arguments would enhance the article's persuasiveness.
Response:- The title of Section 8 has been revised to “Emerging Advantages and Transformative Breakthroughs in Nano-medicine” to better reflect the section’s thematic focus. Additionally, relevant experimental studies and clinical trials have been reviewed and incorporated to illustrate real-world applications of these technologies.
Comment-4: Section 9 could be expanded to discuss the current limitations of nanotechnology, including potential obstacles, challenges, and unresolved issues that may impact clinical translation.
Response:- Section 9 has been expanded to include a detailed discussion of the current limitations of nanotechnology, covering key translational obstacles such as toxicity concerns, immunogenicity, manufacturing complexities, regulatory challenges, and unresolved issues affecting clinical implementation.
Reviewer 2 Report
Comments and Suggestions for Authors
The manuscript delivers a comprehensive, well-curated landscape review of the expanding role of nanomedicine across cancer and autoimmune immunotherapy. The authors clearly invested in synthesizing a broad spectrum of mechanisms, platforms, and translational touchpoints, and the narrative demonstrates an ambitious attempt to bridge foundational immunology with next-generation nano-engineering. However, several mechanistic, structural, and presentation gaps need refinement to strengthen confidence in the conclusions and elevate the manuscript to publication standards.
Major Comments
- The manuscript needs a clearer explanation of what makes the highlighted nanomedicine strategies truly new compared with earlier approaches. Strengthening this contrast will help readers see the real advancement.
- The mechanistic sections are informative but often broad. Adding a few concrete examples of specific nanoparticle types or designs that drive the described immune effects would make the explanations more convincing.
- Figures are visually useful but not well integrated into the text. Please reference them more directly and explain what each figure contributes to the argument.
- Discussion of safety and translational challenges is brief. Expanding on potential toxicity, immunogenicity, and regulatory considerations would provide a more balanced and complete picture.
- Some disease areas, especially in the autoimmune section, feel less detailed than others. Adding more specificity or consistency across diseases will improve clarity.
Minor Comments
- Several sections would be clearer with tighter cross-referencing to figures and tables to improve navigability.
- Terminology should be used consistently for example, “nanocarrier,” “nanoparticle,” and “nanoplatform” appear interchangeably.
- Some unit formatting and hyphenation need correction (e.g., “µM,” “dose-dependent”).
- Please define baseline values or assumptions when describing cytokine shifts or immune signatures in the disease examples.
Author Response
Comment-: The manuscript needs a clearer explanation of what makes the highlighted nanomedicine strategies truly new compared with earlier approaches. Strengthening this contrast will help readers see the real advancement.
Response: - The manuscript has been revised to clearly distinguish how the highlighted nanomedicine strategies differ from earlier approaches.
Comment-: The mechanistic sections are informative but often broad. Adding a few concrete examples of specific nanoparticle types or designs that drive the described immune effects would make the explanations more convincing.
Response:- The mechanistic sections have been strengthened by adding specific examples of nanoparticle designs—including representative lipid nanoparticles, polymeric carriers, and biomimetic platforms—that directly illustrate the immune effects described.
Comment-: Figures are visually useful but not well integrated into the text. Please reference them more directly and explain what each figure contributes to the argument.
Response:- All figures have now been directly referenced within the relevant sections, and brief explanatory descriptions have been added to clarify how each figure supports and strengthens the corresponding arguments.
Comment-: Discussion of safety and translational challenges is brief. Expanding on potential toxicity, immunogenicity, and regulatory considerations would provide a more balanced and complete picture.
Response:- The discussion has been expanded to include a more detailed analysis of toxicity profiles, immunogenicity concerns, biodistribution issues, manufacturing constraints, and key regulatory requirements.
Comment-: Some disease areas, especially in the autoimmune section, feel less detailed than others. Adding more specificity or consistency across diseases will improve clarity.
Response:- The autoimmune section has been updated to ensure greater specificity and consistency across disease examples. Additional mechanistic details, therapeutic contexts, and nanotechnology applications have been integrated uniformly across the discussed conditions, resulting in a clearer and more balanced presentation.
Comment: Several sections would be clearer with tighter cross-referencing to figures and tables to improve navigability.
Response:- Cross-referencing has been improved throughout the manuscript by adding clear links to relevant figures and tables within the text.
Comment: Terminology should be used consistently for example, “nanocarrier,” “nanoparticle,” and “nanoplatform” appear interchangeably.
Response:- Terminology has now been standardized across the manuscript, ensuring uniform and precise use of terms such as “nanocarrier,” “nanoparticle,” and “nanoplatform.”
Comment: Some unit formatting and hyphenation need correction (e.g., “µM,” “dose-dependent”).
Response:- All unit formats and hyphenation inconsistencies have been thoroughly corrected in accordance with standard scientific style guidelines, ensuring accuracy and uniformity throughout the manuscript.
Comment: Please define baseline values or assumptions when describing cytokine shifts or immune signatures in the disease examples.
Response:- Baseline values and underlying assumptions for the described cytokine shifts and immune signatures have now been clearly defined in the relevant sections.
Reviewer 3 Report
Comments and Suggestions for Authors
1] Grammatical Errors and Awkward Phrasing: Numerous issues hinder readability. E.g., Abstract: "radical agent of change in the game of immunotherapy" is cliché and imprecise; "working on a scale both spatial and temporal in such a way that it helps overcome the drawbacks" lacks parallelism and flow. Page 4: "Recent have demonstrated" (missing subject); "Pd-1" (should be PD-1).
2] Discuss failures (e.g., NP immunogenicity in autoimmunity) and ethical issues (e.g., equity in personalized therapies).
3] Repetition and Redundancy: Phrases like "fine-tune immune response" repeat across sections (e.g., Intro, 2.1, 3.1). Section 2.3's cytokine discussion overlaps with 2.1's TME cytokines without adding value.
Author Response
Comment: Grammatical Errors and Awkward Phrasing: Numerous issues hinder readability. E.g., Abstract: "radical agent of change in the game of immunotherapy" is cliché and imprecise; "working on a scale both spatial and temporal in such a way that it helps overcome the drawbacks" lacks parallelism and flow. Page 4: "Recent have demonstrated" (missing subject); "Pd-1" (should be PD-1).
Response:- The manuscript has been carefully edited for grammar, clarity, and parallelism. Clichés and awkward phrasing have been revised, typographical errors corrected (e.g., “Pd-1” → “PD-1”), and sentence structures improved to enhance overall readability.
Comment: Repetition and Redundancy: Phrases like "fine-tune immune response" repeat across sections (e.g., Intro, 2.1, 3.1). Section 2.3's cytokine discussion overlaps with 2.1's TME cytokines without adding value.
Response:- Repetitive phrases and overlapping content, particularly in cytokine discussions, have been streamlined to eliminate redundancy and improve readability.
Round 2
Reviewer 2 Report
Comments and Suggestions for Authors
In the revised version, authors have largely addressed the issues raised by the reviewer. The reviewer has no further comments.